# Bee Pollen and Bread as a Super-Food: A Comparative Review of Their Metabolome Composition and Quality Assessment in the Context of Best Recovery Conditions

**DOI:** 10.3390/molecules28020715

**Published:** 2023-01-11

**Authors:** Mostafa H. Baky, Mostafa B. Abouelela, Kai Wang, Mohamed A. Farag

**Affiliations:** 1Pharmacognosy Department, College of Pharmacy, Egyptian Russian University, Badr City P.O. Box 11829, Egypt; 2Institute of Apicultural Research, Chinese Academy of Agricultural Sciences, Beijing 100093, China; 3Pharmacognosy Department, College of Pharmacy, Cairo University, Kasr el Aini St., Cairo P.O. Box 11562, Egypt

**Keywords:** nutraceuticals, bee pollen, bee bread, phytonutrients, optimized extraction, analysis

## Abstract

Recently, functional foods have been a subject of great interest in dietetics owing not only to their nutritional value but rather their myriad of health benefits. Moreover, an increase in consumers’ demands for such valuable foods warrants the development in not only production but rather tools of quality and nutrient assessment. Bee products, viz., pollen (BP) and bread, are normally harvested from the flowering plants with the aid of bees. BP is further subjected to a fermentation process in bee hives to produce the more valuable and bioavailable BB. Owing to their nutritional and medicinal properties, bee products are considered as an important food supplements rich in macro-, micro-, and phytonutrients. Bee products are rich in carbohydrates, amino acids, vitamins, fatty acids, and minerals in addition to a myriad of phytonutrients such as phenolic compounds, anthocyanins, volatiles, and carotenoids. Moreover, unsaturated fatty acids (USFAs) of improved lipid profile such as linoleic, linolenic, and oleic were identified in BP and BB. This work aims to present a holistic overview of BP and BB in the context of their composition and analysis, and to highlight optimized extraction techniques to maximize their value and future applications in nutraceuticals.

## 1. Introduction

Bee pollen (BP) is the natural pollen of flowering plants collected mainly by honeybees and is considered as the major source of feeding for bee growth [1]. Owing to its nutritional and medicinal value, BP plays pivotal role in human health [2]. Regular BP supplementation can induce several health benefits such as reducing capillary fragility, improving the cardiovascular system, maintaining gut functions, and promoting skin health [3]. Additionally, because of its anti-inflammatory activity along with its anti-androgen effect, BP is used in the treatment of chronic prostatitis [1]. The nutritional and medicinal value of BP is attributed to its richness in several macro- and micronutrients such as carbohydrates [3], proteins, vitamins, amino acids, minerals, lipids, flavonoids, phenolic compounds, glucosinlolates (GLSs), and essential oils [4]. BP is considered as a rich source of certain essential amino acids including arginine, histidine, lysine, tryptophan, phenylalanine, methionine, threonine, leucine, isoleucine, and valine [5]. Additionally, BP contains vitamins, macroelements such as phosphorus, sodium, calcium, magnesium, and potassium, and microelements including copper, manganese, iron, zinc, and selenium [6].

In contrast, bee bread [7] is produced via the fermentation of pollen, honey, and bee digestive enzymes in the presence of lactic acid bacteria in wax-sealed honeycombs [8]. Such fermented BB is considered to be more bioavailable than crude BP which is collected directly from the legs of the bees at the hive entrance with the aid of traps [9]. Owing to their richness in polyphenols and flavonoids, both BP and BB are considered as potent antioxidants and possess several human health benefits [10] including antitumor, anti-inflammatory and antimicrobial effects [11]. The total phenolic content (TPC) of both BP and BB collected from different locations varies between 10.5 and 24.6 mgGAE/g [12] and 2.5 and 37.1 mgGAE/g, respectively, and total flavonoid content (TFC) is reported to be from 1.9 to 4.5 mgQE/g [8]. Compared to BP, fermented BB is different in chemical composition as it contains higher levels of reducing sugars, vitamin K, and is considered as a rich source of polyunsaturated fatty acid (PUFA) that is essential for human health and improving lipid profile [13]. Moreover, both BP and BB have high nutritional value as they are considered as a rich source of fatty acids. BP is rich in long chain fatty acids such as linoleic, α-linolenic, and α-palmitic acid [14], whereas BB is richer in linoleic, oleic, and 11,14,17-eicosatrienoic acids [14]. Although the protein level in BP is higher than that of BB, BB protein is considered more bioavailable and digestible [15]. Recently, BB has been favored as a dietary supplement over BP, which has led to the enlargement of BB production and collection from honeycombs [8]. Standardization of nutraceuticals becomes essential for use to confirm its consistency and safety using different analytical tools [13]. Due to the change in several conditions such as climate, plant origin, collection tools, and collection time, the nutritional and chemical characteristics of bee products are subjected to differences [13]. Therefore, determining the exact chemical composition of such valuable components is important for an estimation of their quality. This study aims to present a holistic comparative overview on the chemical composition of both BP and BB along with extraction and analytical techniques for their major phytochemical classes to ensure their quality and the highest recovery of targeted chemicals.

## 2. Macronutrients

Typical macronutrients identified in BP and BB include carbohydrates, fats, and proteins to be discussed in the next subsections (Table 1).

### 2.1. Carbohydrates

Carbohydrates, some of the major constituents of BP and BB, are derived from the flower nectar. Sugars and sugar alcohols constitute the major part of pollen carbohydrate content (Figure 1) [4]. Fructose, glucose, and sucrose are the major sugars reported in bee products along with other minor sugars including arabinose, isomaltose, melibiose, melezitose, ribose, trehalose, and turanose [49]. Five bee pollen samples collected from China, Romania, and Spain were investigated for their sugar content, viz., glucose, fructose, and sucrose using high performance thin layer chromatography (HPTLC) and high-performance ligand exchange chromatography with pulsed amperometric detection (HPLEC-PAD). Results revealed the presence of sugars in all the samples, including stachyose, sucrose, glucose, galactose, and fructose. Moreover, the quantification of sugars in pollen samples using HPLEC-PAD analysis on a Ca-ligand exchange column revealed that fructose was the major sugar quantified at a level ranging from 15.9–19.9%*w*/*w*, followed by sucrose (15.8–17.2%) and glucose (8.2–1%) [4]. Stachyose, a tri-saccharide, was detected at trace levels in all samples analyzed using HPLEC coupled with MS [4]. The ethanol extract of BP from three different Malaysian species, *Trigona apicalis* L. (Tetrigona), *T. itama* L. (Heterotrigona), and *T. thoracica* L. (Geniotrigona) was examined using GC-MS analysis. Major sugars were identified as mannitol, fructose, xylitol, glucose, ribitol, myo-inositol, and inositol, with an abundance of mannitol in all species at levels of 54.34% in *T. thoracica*, 39.1% in *T. apicalis*, and 33.05% in *T. itama* [16]. These results suggest that the BP sugar profile is low calorie as sugar alcohols do not increase blood sugar levels, and additionally avoid dental caries, making it a good sugar source for diabetic patients [16].

### 2.2. Fats

Dietary fatty acids play a major role in the composition of the cell membrane, are directly related to certain cardiovascular problems, and help in maintaining mental health [50]. The high ratio of unsaturated fatty acids to saturated ones (USFA/SFA) in food plays a pivotal role in the prevention of certain diseases [51]. The fatty acid (FAs) content of 14 Brazilian pollen samples was analyzed using GC-FID, revealing that the unsaturated/saturated fatty acid ratios in unifloral, multifloral, and bifloral pollen ranged between 0.6–1.0, 0.4–0.8, and 0.3–1.7, respectively [17]. Additionally, the most abundant fatty acids detected were oleic acid, linoleic acid, and arachidic acid at levels of 3.9–9.9%, 8.9–49.7%, and 8.9–42.7%, respectively [17]. The profile of FAs could aid in determining the authenticity of BP, whether mono- or multifloral, as in the case of Manuka honey of premium quality which has yet to be determined. The FA profile of 20 Chinese monofloral BP species was studied using GC-MS coupled with selected ion monitoring for an improved sensitivity level (SIM). Results revealed that the total fatty acids (TFAs) from all BP samples ranged from 5 to 10.6 mg/g, and the highest level was recorded in dandelion pollen at 10.63 mg/g, followed by corn poppy pollen with 10.1 mg/g. Among the FAs detected, 14 saturated fatty acids, four monounsaturated fatty acids, and six polyunsaturated fatty acids were identified in the BP sample [18]. Unsaturated fatty acid (USFA) content was detected at a much higher level owing to the abundance of α-linolenic acid, nervonic acid, linoleic acid, and arachidonic acid at average levels of 1.5 mg/g, 0.2 mg/g, 0.08 mg/g, and 0.1 mg/g, respectively, compared to saturated fatty acid (SFA) content [18]. In contrast, palmitic acid was detected as the major SFA in all samples at ca. 1.03 mg/g [18]. The predominance of such USFAs presents an added nutritional and health value to BP. Five fresh Tuscan BP species were investigated for their fatty acid profile using GC. Results revealed the identification and quantification of 18 fatty acids in the bee-pollen samples, 10 of which were saturated and eight were unsaturated [20]. Saturated fatty acids were detected in all examined BP species, including tridecanoic (26.3 g/kg to 50.6 g/kg), palmitic (69.3 to 152.9 g/kg), margaric (255.0 g/kg), and stearic acid (23.9 to 65.4 g/kg). Additionally, all studied pollen species revealed the presence of monounsaturated fatty acids, viz., linoleic, and α-linolenic acid at 55.4 g/kg and 147 g/kg of total fatty acids, respectively [20]. Moreover, polyunsaturated fatty acids (n-6/n-3 PUFA) were detected in all samples at a ratio of 1.70 comparable to that reported in honey fatty acid composition [52]. Owing to the importance of the n-6/n-3 polyunsaturated fatty acid ratio and linoleic acid/α-linolenic acid for human health, Tuscan organic BP is considered a premium source with high nutritional value [20]. HPLC analysis of fresh BP from five different botanical sources revealed its richness in fatty acids. BP samples from monofloral pollen loads from summer squash (*Cucurbita pepo* Thunb.), date palm (*Phoenix dactylifera* L.), sunflower (*Helianthus annuus* L.), rape (*Brassica napus* L.), and alfalfa (*Medicago sativa* L.) (from Saudi Arabia in Al-Ahsa) were analyzed [21]. Results revealed that the highest level of essential fatty acids was found in sunflower (36.1%), whereas date palm contained the least (11.5%) [21]. The amount of various FAs in BP included oleic acid (14.2% in summer squash to 49.0% in alfalfa) followed by palmitic acid, from 11.2% in summer squash to 30.1% in sunflower; linolenic acid, from 8.7% in date palm to 29.8% in sunflower; stearic acid, from 4.7% in date palm to 29.2% in summer squash; and linoleic acid, from 4.9% in date palm to 27.4% in sunflower [21]. These results suggest major differences in FA composition in BP according to its origin [21], which has yet to be examined from other several locations worldwide. This variation of FA composition among the samples was in accordance with other reports which revealed that differences in FA levels mainly depends on the source, geographical origin, age, and nutritional status of plants required for the development and brooding of honey bees [21,52,53]. In another study, Lithuanian BP and BB samples were analyzed for their FA composition using GC-FID [22], with the identification of 22 versus 21 fatty acids in bee pollen and bee bread, respectively [22]. Results showed variation in total saturated fatty acids (TSFA) ranging from 0.51 to 1.8% in bee products. Meanwhile, total unsaturated fatty acid (TUSFA) ranged from 0.03 to 3.7% in bee samples [22]. The highest yield of fatty acids was obtained by heating at 70 °C for 90 min. Moreover, the optimal conditions (temperature and extraction time) revealed excellent linearity with coefficients of r^2^ > 0.9998 for fatty acid determination. Additionally, the limit of detection (LODs) and limit of detection quantification (LOQs) ranged from 0.2 to 0.5 g/mL and 0.6 to 1.6 g/mL, respectively. The developed approach was validated for the quantification of fatty acids in bee products with great precision and to be employed for other resources [22].

The ethanol extract of three Malaysian BP samples was analyzed using GC-MS revealing seven fatty acids from the three species (*T. apicalis*, *T*. *itama*, and *T. thoracica*). α-Linolenic acid showed the highest level, followed by linoleic acid detected at 0.07–1.11% in all bee samples [16]. In another study, the non-polar extract of BP from *Cocos nucifera* was investigated using GC-MS [23] with the major fatty acids detected as palmitic acid, oleic acid, and behenic acid. *β*-Sitosterol and stigmastan-3,5-diene were the major phytosterols alongside ergosta-5,24(28)-dien-3-ol, campesterol, sitosterol, and fucosterol, though at much lower levels [23]. The main active constituents of 12 Chinese varieties of monofloral BP were analyzed using GC-FID [24], revealing that the major fatty acids were α-linolenic acid (25.1%), palmitic acid (19.6%), oleic acid (17.3%), linoleic acid (8.7%), and stearic acid (2.9%) [24]. Additionally, only *Nelumbo nucifera* Gaertn. (NNG) pollen showed the absence of α-linolenic acid, whereas all specimens contained palmitic acid, oleic acid, linoleic acid, and arachic acid. The total unsaturated/total saturated fatty acid ratio was greater than 1.0 in all samples except NNG pollen [24]. Such higher ratios of TUS/TS in pollen add to its value to be used as a dietary food supplement, as a hypocholesterolimic agent, and to reduce cardiovascular disease risk [24]. The considerable variations in the unsaturated/saturated fatty acid ratio among analyzed samples could be due to variations in certain factors such as botanical origins, processing, and storage conditions [24]. Another study reported the analysis of seven BP using GC-MS, revealing the identification of 11 fatty acids [25], with like unsaturated fatty acids amounting to 69.14%, with α-linolenic (ω-3 FAs) as the major form detected at 36.25% of the total fatty acids, which was more than that previously reported (35.5%) [54] indicating that *F. esculentum* BP represents a potential source of ω-3 FAs among the examined BP [25]. Dietary ω-3 FA intake is beneficial to human health by reducing blood lipids, and lowering the risk of cardiovascular diseases [25]. In another study, the FA content of eight BB samples from various plant sources were analyzed using GC [19]. Results revealed the presence of 37 fatty acids with a predominance of palmitic acid (22.3–38.7%), α-linoleic acid (6.3–37%), oleic acid (3.9–21.2%), stearic acid (1.3 to 6.3%), and linolenic acid (0.2–40.7%). Among the analyzed samples, the highest level of omega-3 FAs was detected in the cotton BB sample at a level of 41.3%. Moreover, the USFA/SFA ratio in BB ranged between 1.38 and 2.39, making it a potential source for unsaturated fatty acids [19].

Fatty acid content in BP and BB samples from the same bee hive collected from five different origins in Turkey was determined using GC–MS [26]. The total identified fatty acids ranged from 60.2 to 86.5% with a total of 22 different fatty acids, including palmitic acid, stearic acid, oleic acid, linoleic acid, eicosenoic acid, and linoleic acid [26].

Collectively, BP and BB samples are considered as essential nutrients, being rich in FAs, especially unsaturated fatty acids such as oleic acid (3.9–21.2%), linoleic acid (6.3–49.7%), and linoleic acid (0.2–40.7%), with myriad health benefits and important for heart health, though with differences based on origins.

### 2.3. Proteins

Amino acids (AAs) are considered as essential nutrients as they play a pivotal role in growth, nitrogen balance for humans [55], protein synthesis, regulating gene expression, cell signaling, oxidative defense capacity, intestinal health, and immune response [56]. They are typically classified as both essential and nonessential according to their importance or the ability of the human body to synthesize them. The abundance of AA content in BP and BB plays a vital role in the regulation of several physiological and biological activities even at trace levels (Table 2). HPLC-fluorometric assay was used to identify both free and total amino acids in 32 Spanish bee-pollen samples from *Cistus ladanifer* L. and *Echium plantagineum* L. [28]. Detection ranged from 0.4 to 29 pmol (picomol), while reproducibility and recoveries were at 20.4% and 78.8%, respectively. Results revealed the presence of 22 free amino acids in BP, with proline detected as the major amino acid at ca. 20.27 mg/g, followed by leucine, alanine, phenylalanine, histidine, serine, and glutamine in the range of 0.6–0.9 mg/g; *C. ladanifer* was also the major source of FAA content [28]. With respect to total amino acids, proline was the major AA detected at 22.88 mg/g, followed by glutamic acid (17.88 mg/g), aspartic acid (15.10 mg/g), lysine (10.97 mg/g), and leucine (10.81 mg/g) [28]. Proline was also the major AA in honey with average contents of 800–850 mg/kg, which nearly equaled half of the free amino acid content of that in honey [28]. The higher content of proline in bee products is a good indicator of its sensory characteristics and freshness. In another study using a new, sensitive labeling reagent, 2-(11Hbenzo[α]-carbazol-11-yl) ethyl chloroformate, for derivatizing rape BP, amino acids coupled to fluorescence detection [27] were used for the quantitative analysis of amino acids. Results revealed the identification of 19 AAs in rape derived BP samples, and Arg, Asp, Glu, and Pro were detected at higher levels with detection limits ranging from 7.2 fmol for tryptophan to 8.4 fmol. Results showed that the new reagent has a high detection level and excellent fluorescence properties, especially for cystine and tryptophan, which cannot be detected using other standard labelling reagents [27]. The use of LC-MS/MS to profile AAs in five fresh samples of BP and BB was reported by Bayram et al. [29]. Proline was likewise detected as the major amino acid abundant in all samples. It could be used as a freshness index of BP samples, in addition to its function as a phagostimulatory metabolite and to supply energy for bees’ flights. L-proline was detected in both BP and BB samples at 8.39–16.68 mg/g and 4.94–22.21 mg/g, respectively, followed by asparagine and aspartic acid; a high content of γ-aminobutyric acid (GABA) (2.33–5.08 mg/g) was also detected [29]. In addition, relatively high levels of L-phenylalanine were recorded in both BP and BB samples at comparable levels of 1.3–3.34 mg/g [29]. The ethanolic extract of BP (from three Malaysian species) was examined using the GC-MS technique. The results showed that amino acids in BP samples collected from *Trigona apicalis* were composed of L-alanine, L-valine, and L-asparagine, while L-proline was abundant in *T. thoracica* [16]. The main active constituents of 12 Chinese monofloral BP varieties were analyzed for their AAs [24] revealing that proline, glutamic acid, and aspartic were the major amino acids [24]. BP collected from the *Rhododenron ponticum* plant was analyzed for its amino acid content revealing the identification of 42 amino acids with total free amino detected at 149.46 mg/g, represented by L-asparagine and proline as the major amino acids detected at ca. 62.6 and 14.3 mg/g, respectively [30]. Being rich in both essential and non-essential AAs, BP and BB are considered as important sources of AAs, especially proline (4.9–22.8 mg/g) to fulfill the daily requirements of AAs.

## 3. Micronutrients (Vitamins/Minerals)

### 3.1. Vitamins

Vitamins are an essential group of organic compounds important for both the growth and development of healthy humans and with myriad health benefits [57]. BP is also called a “vitamin bomb” owing to its richness in almost all vitamins. The beneficial effects of vitamins depend on an adequate intake of all vitamins. Several reports demonstrated that BP and BB are rich in vitamins content adequate to fulfill human needs [58]. Bee products (BP and BB) are reported to contain fat-soluble vitamins (A, D, E, and K) and water soluble vitamins (C and B complex) with different levels depending on the botanical origin and time of collection [49]. Vitamin E, mainly tocopherol, a fat-soluble vitamin with antioxidant potential, was reported to be abundant in both BP and BB [59]. A total of 20 BP samples collected from different apiaries from southern Brazil were analyzed for their vitamin content revealing the abundance of α, β, γ, and δ-tocopherol at levels 5–73 μg/g, 1–10 μg/g, 2–12 μg/g, and 1–84 μg/g, respectively [60]. Compared to BP, BB samples from northern Portugal showed tocopherol content at much higher levels, mainly δ-tocopherol (77–293 mg/g), α-tocopherol (1–37 mg/g), and γ-tocopherol (1–35 mg/g) [61]. Moreover, vitamin C, a water-soluble vitamin with potential antioxidant capacity, has been reported at higher levels in BP (14–797 μg/g) and much lower levels in BB (0.06–0.11 μg/g) [57]. Vitamin C plays a pivotal role in amino acids, cholesterol, collagen, and some hormone biosynthesis, aside from its health benefits such as an immunostimulant, anticarcinogenic, and in reducing the risk of cardiovascular disease [62]. The vitamin content of *Rhododendron ponticum* L. BP was analyzed using HPLC-FLD/UV, revealing the detection of vitamins A, B1, B2, B6, E, K1, and K2, and vitamins B5, B7, B12, and C, respectively [30]. Results revealed the abundance of vitamin C at 16240 μg/100 g, followed by B5 (1940 μg/100 g), B2 (735 μg/100 g), E (490 μg/100 g), B6 (455 ug/100 g), B1 (315 μg/100 g), β-carotene (264 μg/100 g), B12 (0.86 μg/100 g), retinol (22 μg/100 g), B7 (46 μg/100 g), and lower levels of K2 (1.51 μg/100 g) [30]. From the aforementioned reports, bee products are considered as a potential source for vitamins to supply the human body and their consumption can fulfill a human’s daily vitamin needs, especially δ-tocopherol (5–293 mg/g), α-tocopherol (1–37 mg/g), γ-tocopherol (1–35 mg/g), and vitamin C (14–797 μg/g).

### 3.2. Macro- and Microelements

Macro- and microelements are essential for growth as they are involved in many biochemical processes and components of biological structures, i.e., bones, nerves, and muscles. Moreover, minerals are important in the function of enzymes, hormones, and pigments, such as oxygen-carrying hemoglobin [31]. Additionally, an assessment of potentially toxic elements with health hazards such as arsenic, cadmium, mercury, and lead is important for ensuring the quality of bee products [63]. Macroelements, viz., Ca, K, Mg, Na, P, and S, and microelements, viz., Co, Cu, Fe, Mn, Mo, Se, and Zn (Table 3) were investigated in Polish bee pollen samples using inductively coupled plasma-mass spectrometry (ICP-MS) and inductively coupled plasma optical emission spectrometry (ICP-OES) techniques [31]. The results revealed the abundance of K, P, and S at high levels (4233.33 mg/kg, 4050.00 mg/kg, and 2383.33 mg/kg, respectively). Moreover, the highest micronutrients level were determined as Fe, Zn, and Mn with concentrations of 114.5 mg/kg, 31.3 mg/kg, and 25.0 mg/kg, respectively [31]. In another study, a new fast and simple analysis method was applied for detecting total amounts of Ca, Cu, Fe, Mg, Mn, and Zn in 10 collected BP samples by combining line source-flame atomic absorption spectrometry (LS-FAAS) with ultrasound-assisted solvent extraction (UAE) [32]. Results showed a range of concentrations from 3.0–4.4% (Ca), 2.2–3.5% (Cu), 2.2–3.9% (Fe), 1.8–3.5% [64], 2.0–2.8% [65], and 2.1–3.1% [19,32]. In order to determine macronutrients and N, C, and S elements of 30 samples of bee pollen, ^13^C nuclear magnetic resonance (NMR) spectroscopy was used [66]. RSEPcal/RSEPval values of 0.3/0.6% for the total of NHCS components, 0.3/0.4% for C, 1.8/1.9% for N, and 4.2/6.1% for S quantification were obtained using ^13^C NMR spectra to model the elemental composition of BP. Macronutrients were measured using partial least squares (PLS) models with RSEPval errors in the 2.2–2.5% range. Results revealed that a single solid-state NMR spectra of powdered BP can be used to assess numerous properties of this complex natural product without the use of solvents or separation processes [66].

The combination of diffuse reflectance spectra in near-infrared region spectroscopy with PLS regression was reported as a faster alternative to standard methodologies employed for mineral quantification. A total of 154 BP samples were analyzed revealing the presence of a wide range of Ca, Mg, Zn, P, and K levels, with inductively coupled plasma optical emission spectrometry (ICP-OES) utilized as a reference method [33]. These results indicated that the NIR-PLS method was useful to quantify Ca (1346 to 3724 mg/kg), K (3182 to 7376 mg/kg), Mg (915 to 1744 mg/kg), P (3257 to 6886 mg/kg), and Zn (38 to 76 mg/kg) from 11 Brazilian States and Federal Districts [33]. Another simple modified analytical method using ICP-MS was used for the detection of 39 major, minor, trace, and rare earth elements using small mass samples of BP. Results showed that all elements exhibited adequate accuracy and reproducibility, with a low degree of detection for trace and rare earth [67]. The mineral content in 12 Chinese varieties of monofloral BP was analyzed using inductively coupled argon plasma atomic emission spectrometry (ICP-AES) [24]. Results showed that macro and microelements were detected in BP samples at high levels especially, viz., P at 5946 mg/kg, K at 5324 mg/kg, Ca at 2068 mg/kg, and Mg at 1449 mg/kg [24]. Inductively coupled plasma mass spectrometry (ICP-MS) was used to determine element levels in Turkish BP and BB samples collected from five different locations [26]. Results showed the presence of 42 distinct elements at various levels in all of the BP and BB samples [26]. The five elements found at the highest levels in both BP and BB were K (5429.27–8994.25 mg/kg), P (4221.86–5948.96 mg/kg), Mg (1033.72 mg/kg), Ca (189.69–447.13 mg/kg), and Si (47–537.97 mg/kg) [26]. Another study compared the mineral content in 18 BP and BB samples from Lithuania using ICP-MS. Results revealed that BP was rich in macrominerals such as Ca (997–2455 mg/kg), compared to less than 612 mg/kg in BB, and Mg (644–1004 mg/kg), compared to less than 409 mg/kg in BB, whereas Co (0.011–0.1 mg/kg) and Sr (0.73–5.37 mg/kg) were the major microminerals in BP [34]. In another study, BP of *Rhododenron ponticum* was analyzed using ICP-MS [30] revealing that total element content reached 13,997 mg/kg with K, P, Ca, and Mg as major elements detected at 3002, 9145, 459, 1113 mg/kg, respectively. Recently, macro- and microelements were analyzed in commercial Turkish BP samples revealing that Mg (378.65 mg kg^−1^) was the most abundant element in BP, followed by Fe (50.88 mg kg^−1^). Moreover, BP can supply 20–35% of daily Cu, Mn, and Se requirements while decreasing the content of potentially toxic elements (Co, Cr, Cu, Fe, Mg, Mn, Mo, Ni, Se, Zn, and V) [68]. Bee products are rich in macronutrients and can fulfill the daily requirements of children, adults, and pregnant women with safe sources of macroelements and lower levels of toxic elements.

## 4. Phytonutrients

Several phytochemical groups were reported in BP and BB samples such as volatiles, coenzyme Q10, carotenoids, anthocyanins, phenolics, and glucosinolates (Figure 2), which are discussed in the next subsections.

### 4.1. Volatile Compounds

Volatile compounds of different aroma types are detected in BP as a result of the collection of nectar from flowers by the honey bee [57], and whether they can be used to trace BP flower origin is not reported extensively in the literature. The volatile content in 14 BP samples from Lithuania was analyzed using solid-phase micro-extraction (SPME)-GC-MS technique with a total of 42 volatiles including nonanal 1.5–20.1%, dodecane 1.2–34.6%, and tridecane 1.4–24.7% [35]. Likewise, the volatile content of three different polyfloral Lithuanian BP samples was evaluated using SPME-GC-MS. Results revealed that styrene was the most abundant, accounting for 19.6–27.0%. Other volatiles that appear more flower-specific included limonene, hexanal, nonanal, and 1-tridecene (43.3%) [36]. The volatile oil content of three stingless BP samples (genus *Scaptotrigona* sp.) collected from mid-north region of Brazil was determined using GC-MS. Results revealed the identification of 41 volatile compounds belonging to different classes with an abundance of hydrocarbons and esters; the most frequent volatiles among the analyzed samples were kaur-16-ene, methyl cinnamate, benzyl acetate, methyl benzoate, methyl hydrocinnamate, and ethyl phenylacetate [38]. Compared to other studies, hydrocarbons and esters were reported to be abundant in stingless BP samples. The volatile constituents in ground BP and its aqueous solution (1:1 *w*/*v*) were investigated using HS-SPME/GC–MS analysis [37]. A total of 25 compounds were identified in ground bee pollen mostly belonging to aldehydes, such as 2-methylbutanal, 3-methylbutanal, pentanal, hexanal, (E)-2-hexenal, heptanal, octanal, and nonanal. Meanwhile, ground BP aqueous solution showed the presence of 22 compounds, of which four new volatiles were identified as 2-methyl-propanal, hexanoic acid methyl ester, 2-heptanone, and 2-octene [37]. The volatile compounds detected in BP and BB are related to the floral source from which the bee products are collected [38]. The abundance of volatile constituents in bee products can be used as an indication of their freshness and further adds to the sensory characteristics of commercial products.

### 4.2. Coenzyme Q10

Coenzyme Q10 (CO-Q10) plays a fundamental role in the mitochondrial electron transport chain and as an antioxidant in plasma membranes and lipoproteins [3]. The COQ-10 concentration in BP was determined using accelerated solvent extraction (ASE) techniques such as HPLC-DAD, and 11 commercial samples, including rape bee pollen, tea bee pollen, apricot bee pollen, and mixed bee pollen, were analyzed for their CO-Q10 content [1]. The results revealed that extraction temperature had a considerable effect on extraction yield, where adjusting the temperature by more than 80 °C provided satisfactory extraction yields. The assay was well validated and found linear over a concentration range of 0.25–200 mg/L, and LOD and LOQ and were detected at 0.16 and 0.35 mg/kg, respectively. The percentage of recoveries were above 90% with below 6.3% inter- and intra-day precision [1]. In another study, the CO-Q10 amount in BB samples from Poland was determined using LC/MS-MS and detected at 11.5 μg/g [39].

### 4.3. Carotenoids

The carotenoid composition of bee-pollen samples were analyzed by Salazar-González et al. using rapid resolution liquid chromatography (RRLC) coupled to UV–Vis spectrophotometry and Digital Image Analysis (DIA) [40]. Results led to the identification of *α*-tocopherol (4.7–95.9 μg/g) along with 10 other carotenoids, including phytoene (0.12–17.82 μg/g), lutein isomer 1 (1.62–131.03 μg/g), lutein isomer 2 (1.3–137.5 μg/g), anteraxanthin isomers 1 (1.6–131.0 μg/g), zeaxanthin (from 12.8 to 256.4 μg/g), zeinoxanthin (10.5–996.1 μg/g), lutein (0.99–4.6 μg/g), β-cryptoxanthin (2.1–16.3 μg/g), and β-carotene (0.68–3.6 μg/g). The correlation of colorimetric coordinates with carotenoid level was further estimated using multiple linear regression (MLR) [40]. In another study, 16 chestnut-derived BP samples collected from two regions in Turkey were analyzed for carotenoid level using HPLC-DAD [41]. Results showed the presence of five carotenoids, identified as lutein (3.8 to 33.4 mg/kg), zeaxanthin (4.8 to 36.3 mg/kg), β–cryptoxanthin (7.5 to 44.6 mg/kg), β-carotene (7.9 to 152.4 mg/kg), and *β*-carotene (from 2.4 to 395.0 mg/kg) [41]. By comparing carotenoid levels in BP to that of carrot (a major source carotenoids), the abundance of carotenoid in BP at higher levels was revealed, posing it as an essential carotenoids source [41].

### 4.4. Glucosinolates (GLS)

Glucosinolates are secondary plant metabolites with potential health benefits and their quality and number differ among different bee pollens depending on plant source. Sulforaphane (SFN) is an isothiocyanate derivative produced upon the reaction of enzyme myrosinase with glucosinolate, typically upon cell crushing [69]. SFN was identified for the first time in five bee pollen samples using the LC-MS/MS method [42]. SFNs were detected at trace levels in some bee-pollen samples (<23 g/kg). The average analyte recovery rate ranged between 92 and 106% in all cases [42], with LC-MS/MS found selective, linear from 8 to 1000 g/kg, with both relative standard deviation (percent RSD) and relative error (percent RE) values less than 10%, and both LOD and LOQ limits at 3 g/kg and 8 g/kg, respectively [42]. GLS content in 49 BP samples from four different apiaries in Spain was determined using UPLC-Q-TOF/MS, revealing variation in GLS levels ranging from 34 to 9806 μg/kg [43].

### 4.5. Phenolics

Phenolics represent a group of phytochemicals found in almost all plants and are of potential value to human health. Phenolic acids and phenolics showed higher ability to activate enzymes of antioxidant protection in cells, which can prevent oxidative stress damage in cells [70]. Several phenolics belonging to flavonoids (listed in Appendix A) and phenolic acids (Appendix A) were reported in BP and BB. Total phenolic–flavonoid content (TPC-TFC) in BP and BB samples from the same bee hive from five different locations in Turkey [26] was assessed with TPC ranging from 8.2 to 43.4 mgGAE/g, whereas TFC ranged from 1.8 to 4.4 mgQE/g [26]. Moreover, BP showed much higher TPC levels than BB samples mostly affected by the extraction method, extraction solvents, and botanical origins of the samples, which was in accordance with several studies [12,71]. The polyphenolic content of 35 BP samples collected from different locations in India was investigated using UHPLC-DAD-MS/MS, with 60 identified compounds, including 38 flavonoids mostly comprising the flavonol subclass, 21 phenolic acids, and one glucosinolate [47]. On the other hand, 13 hydroxycinnamic acids derivatives, 4 hydroxybenzoic acids, and 4 phenolic glycerides were quantified as follows: catechin, 0.94–19.1 mg/100 g; rutin, 4.81–24.8 mg/100 g; quercetin, 3.14–15.9 mg/100 g; luteolin, 1.06–5.8 mg/100 g; kaempferol, 0.12–9.3 mg/100 g; and apigenin, 0.46–3.02 mg/100 g [47]. Likewise, nine polyphenols were identified from BP samples collected from Greece (Viannos, Crete), quantified as follows: ferulic acid, 149.1 µg/g; o-, *p*-coumaric acid, 36.6 µg/g; quercetin, 29.6 µg/g; cinnamic acid, 23.4 µg/g; naringenin, 21.9 µg/g; hesperitin, 3.0 µg/g; and kaempferol, 7.83 µg/g, using nano-liquid chromatography system [44]. The optimized analytical method was validated and the intra-day and inter-day RSD % for retention duration, retention factor, and peak area repeatability were below 4.68 and 5.57%, respectively [44]. Compared to traditional HPLC methods, the new method runs in nanoflow conditions, resulting in higher sensitivity and shorter analysis times [44]. The phenolic content from a BP methanol extract sample using HPLC analysis was investigated [45]. 3, 4-Dimethoxycinnamic acid (45.8 mg/mL) was identified as the most abundant phenolic, alongside gallic acid, catechin, and quercitin as major phenolics [45]. Flavonoid and phenolic acid contents in a commercial BP sample collected from local markets in Valladolid, Spain, were investigated using supercritical fluid chromatography (SFC) [46]. Nine phenolic compounds were identified and quantified as follows: catechin, 22.15 mg/kg; quercetin, 22.02 mg/kg; *p*-coumaric acid, 11.62; and cinnamic acid, 3.70 mg/kg [46]. Furthermore, the best results were obtained at LODs and LOQs less than 5 microg/mL; however, the RSD percentage values for method repeatability and inter-day reproducibility were less than 3% and 10%, respectively [46].

In another study, phenolic compounds in 16 chestnut BP samples from two different regions in Turkey were analyzed using HPLC-DAD [41]. Major phenolics included luteolin (from 0.3 to 0.9 mg/g), hyperoside (0.18 to 1.1 mg/g), vitexin (0.354 to 1.720 mg/g), chalcone (0.01 to 0.05 mg/g), rosmarinic acid (2.0 to 11.5 mg/g), pinocembrin (0.06 to 0.1 mg/g), and chrysin (0.01 to 0.04 mg/g). It could be deduced that BP richness in phenolics presents a good source of antioxidants for human health [41]. Five fresh samples of BP and BB were analyzed using LC-MS/MS [29] with phenolics detected at much higher levels in BP over BB, exemplified by caffeic acid (12.4–56.1 mg/100 g), ethyl gallate (0.04–3.1 mg/100 g), trans-ferulic acid (23.2–107.6 mg/100 g), and myricetin (20.4–244.7 mg/100 g) [29]. In contrast, other phenolic acids were found at higher levels in BB versus BP, such as protocatechuic acid (166.61 mg/100 g), *p*-coumaric acid (28.7–142.4 mg/100 g), quercetin (381–3918 mg/100 g), isorhamnetin (1227 mg/100 g), luteolin (21.9–3490 mg/100 g), salicylic acid (19.2–65.2 mg/100 g), chlorogenic acid (3.89–36.09 mg/100 g), 2,5-dihydroxybenzoic acid (2.69–35.09 mg/100 g), kaempferol (112.94–2681.20 mg/100 g), and gallic acid (34.65–347.37 mg/100 g) [29].

A mixture of pollen samples from different plant species collected from Bayburt, Turkey, were investigated for its phenolic and fatty acid content [48]. A total of 23 phenolics were quantified, represented by rutin (115.4 mg/kg) followed by kaempferol (9.8 mg/kg), quercetin (7.8 mg/kg), myricetin (2.2 mg/kg), and p-coumaric acid (0.5 mg/kg) [48].

Phenolic acid and flavonoid content from Lithuanian BP samples were detected using an HPLC-electrochemical detector (ECD) system [35], with major forms detected in all samples, including 2-hydroxycinnamic acid (43.4–179.9 μg/g), rutin (156.2–955.7 μg/g), and quercetin (24.0–529.8 μg/g) [35]. The polar extract of BP from *C. nucifera* was analyzed using HPLC-ESI-MS/MS [23]. Results identified flavonoid glycosides, such as isorhamnetin-3-O-(2″-O-rhamnosyl) glucoside, isorhamnetin-3-O-(2″,3″-O-dirhamnosyl) glucoside, isorhamnetin-di-3,7-O-glucoside, quercetin-3-O-rhamnosylglucoside, quercetrin, and isorhamnetin-3-O-(2″-O-rhamnosylacetyl) glucoside, along with hydroxycinnamic acid amide derivatives [23]. In another study, 33 phenolics were detected in *Rhododendron ponticum* L. BP using LC-MS/MS [30]. Among the detected compounds, myricetin was detected at the highest level (3744.3 μg/100 g), followed by epicatechin (1350.4 μg/100 g), catechin (1207.1 μg/100 g), and tyrosol (1137.6 μg/100 g) [30]. Differences in flavonoid compositions among studies are likely attributed to the BP floral source among other factors. The abundance of phenolics and flavonoids with anti-oxidant potential in bee products indicates its potential role in protecting human cells from oxidative stress adding to its health benefits. 

### 4.6. Anthocyanins

Anthocyanins are colored water-soluble pigments belonging to flavonoids that impart color to fruits and flowers, aside from their several biological activities such as the treatment of neurodegenerative and vascular diseases due to its antioxidant activity [2]. Anthocyanin composition in five Spanish dark blue BP samples was studied using HPLC/PDA/MS. Results revealed that the anthocyanin content ranged from 45 to 80 mg/100 g of blue pollen, which was represented by delphinidin, cyanidin and petunidin-3-O-glucoside, delphinidin, cyanidin, peonidin and malvidin-3-O-rutinoside, and cyanidin-3-(6-malonylglucoside), with the major one being petunidin-3-O-rutinoside [2]. The pigments derived from the blue-black BP were compared to pigments collected from *Fuchsia extorticata* pollen using HPLC [7], identified as delphindin, petunidin, and malvidin-3-O-glucosides, as well as delphinidin-3-O-glucosides [7]. The flavonol glycosides kaempferol-3-sophoroside, quercetin-3-sophoroside, and kaempferol 3-neohesperidoside were detected alongside anthocyanins and confirmed using ^1^HNMR spectroscopy [7].

## 5. Processing and Extraction Methods of BP Bioactives

BP should be typically placed in a sealed container or bag after harvesting to kill mites and then stored in a freezer at −18 °C for at least 48 h [72]. Prior to freezing the pollen, it should be immediately cleaned to remove major pollutants such as dead bees and ants. Leading pollen producers often choose to use large boxes with trays inside to keep their harvests free of large debris, such as dead bees. Pollen from bees degrades over time in terms of its bioactivity, and further nutritional and functional properties are influenced by processing conditions applied to fresh pollen before storage [73] suggesting that BP is best consumed frozen, similar to royal jelly.

Bee bread needs to be gathered from honeycomb and undergoes drying and cooling, followed by segmentation and separation from beeswax [74,75]. Different technologies have been developed for the separation of the beebread, such as vacuum drying, water soaking or freezing, segmentation, and the filtering method. It is recommended that modern non-destructive methods such as acoustic drying should be applied for bee-bread extraction. Moreover, as bee bread is the hive product which undergoes solid-state fermentation from freshly collected bee pollen and is stored in the hive for long period, bee bread has less chance to be spoiled than fresh bee pollen [76]. Nevertheless, poor storage conditions also deteriorates bee pollen and bee-bread quality. It is hypothesized that when taken directly from the freezer, without being dried, bee pollen has a superior taste and consistency. Fresh pollen has a smooth, soft texture, which makes eating it pleasant. The pollen of bees has a high moisture content, which requires a dehydration process (artificial drying) to prevent against rapid fermentation and spoilage [77]. However, when the bee pollen is dried, its taste changes to a large extent. Among all organoleptic features, color and flavor are most affected by storage and conservation conditions [78]. A sensory transformation occurs when bee pollen is transferred into bee bread. The acidity from lactic acid during fermentation brightens endowed bee bread with flavor and possible astringency or bitterness [79].

### 5.1. Polysaccharides Extraction

Differences in BP and bee-bread composition were observed in the context of different extraction methods used for its isolation. Typical extraction methods include solvent extraction, including hot water extraction, acid–base assistance extraction, ultrasonic microwave-assisted extraction, enzymatic digestion, and mixed extraction methods (Figure 3) [80,81]. Hot water can be widely applied for polysaccharide extraction currently, as most polysaccharides have desirable solubility. Nevertheless, some high-molecular-weight polysaccharides or acidic polysaccharides have poor solubility in hot water and alkali–water solutions are chosen as an alternative extraction method for BP and bee-bread polysaccharide extraction (Figure 4) [82]. Moreover, hot water extraction and the alcohol-precipitation method were chosen as the most widely used extraction methods with high yields ranging from 8.5 to 35.8% [83,84]. Zhu et al. extracted polysaccharide from *Fagopyrum esculentum* moench. bee pollen using the aforementioned method, and monosaccharides were composed of mostly arabinose, glucose, and galactose, reaching 92.5% of the total monosaccharides [85].

### 5.2. Protein and Peptides

Protein makes up to 14–30% (W/W) of dry pollen with 20 essential amino acids [86]. Maryse Vanderplanck developed a standardized protocol for the extraction of proteins and polypeptides (m.w. >10 kDa) from BP [87]. Phenol/sodium dodecyl sulfate (SDS) combining procedure was used to extract proteins from dry residual pellets, which were further applied for crude protein quantifications based on bicinchoninic acid (BCA) or Bradford assays. Ion-exchange column and gel filtration chromatography methods were used to isolate a reversibly glycosylated polypeptide (RGP) from rape (*Brassica napus* L.) BP [88], which plays important roles in the development of pollen. As BP and BB are rich in proteins, several attempts have been applied to use digestive proteolytic enzymes to obtain bee-pollen/bee-bread protein hydrolysates, which increased the nutritional values and decreased allergic risks. Tanatorn Saisavoey and colleagues hydrolyzed bee-pollen protein using commercially available food-grade enzymes, and different peptide fractions from bee-pollen hydrolysate showed potent anti-inflammatory activity [89]. Bee-pollen protein hydrolysates from Alcalase^®^, Neutrase^®^, and Flavourzyme^®^, showed selectively antiproliferative effects against lung cancer cells, and were confirmed as a rich source of bioactive peptides [90]. A novel angiotensin-I-converting enzyme (ACE) inhibitory peptide was identified from rape bee-pollen protein hydrolysate using alcalase [91]. Three proteases involved in digestion, including pepsin, trysin, and papain, were applied to bee bread and enzymatic hydrolysates were obtained with antioxidant activity and ACE activities [92].

### 5.3. Lipids and Fatty Acids

Total lipid content in bee pollen and bee bread varies from 1–10%, calculated as dry form. Soxhlet extraction and Folch method are two of the most commonly used preparation methods for lipids [93], using petroleum ether [94], ethanol, or chloroform as solvents [95]. Alternatively, using supercritical carbon dioxide, BP can be broken down to extract its lipids [96], providing a more ecofriendly approach being solvent free.

### 5.4. Phenolics

As a natural source of plant metabolites, BP contains rich polyphenols, including phenolic acids and flavonoids, that provide its diverse biological activity. Ethanol extract was mostly prepared from the BP and several different polyphenols were identified indicating that such extracts of bee pollen may be further used as functional/functional food. Comparisons of different extraction solvents, viz., 70% aqueous ethanol, 100% ethanol, and methanol and water, as well as different extraction processes (agitation, maceration, reflux, and sonication), on the extraction of antioxidant phenolics were attempted. Results showed that 70% aqueous ethanol with agitation led to the optimal conditions to maximize the extraction of polyphenols with the strongest anti-oxidant properties. Additionally, prior enzymatic extraction before solvent extraction on BP showed a satisfactory amount of polyphenols. Phenolics were also extracted from bee bread using different procedures. Florina Dranca and colleagues optimized phenolic extraction from bee bread, and using ultrasonic treatment at 64.7 °C and 23 min showed the highest extraction yield [97]. As phenolics presented in bee pollen and bee bread showed in the free and conjugated forms, water/methanol/formic acid (volume ratio: 19.9/80/0.1) was applied for the extraction of total phenolics. The dried extracts were extracted with diethyl ether to yield the free forms and NaOH was added to the pellets, and extracted again with diethyl ether to obtain cell wall bound phenolics. Results showed a higher content of total phenolic and flavonoid contents in the conjugated form of bee pollen and bee bread than the free forms [98]. Moreover, the gamma ray irradiation effect on bee bread has also been applied for disinfection purposes, and a recent study showed that irradiation with 5 kGy and 20 kGy irradiation exerted a positive effect on antiradical capacity, which correlated with increased concentrations of polyphenolics after irradiation [99].

## 6. Conclusions and Future Perspectives

The comparative composition and analysis tools as well as the processing of bee products were discussed herein. Dietetics pays great attention to novel dietary sources based on natural products with both nutritional and health value. BP and BB are rich in both macro- and micronutrients including carbohydrates, lipids, and proteins in addition to vitamins and minerals. Moreover, BP and BB were reported to contain high levels of dietary phytochemicals with medicinal value such as carotenoids, volatiles, phenolic acids, and flavonoids. Several extraction techniques were employed for BP and BB extraction including solvent extraction, acid–base assistance extraction, ultrasonic microwave-assisted extraction, and enzymatic digestion. Moreover, advanced analytical methodologies were employed in the analysis of macro-, micro-, and phytonutrients to ensure quality characteristics. Little is known on how the fermentation process affects BP chemicals to yield BB, especially with regard to phenolics as the major components of such bee products. Being nutritionally and medicinally important, studies linking pharmacological activities with mechanisms of action are still needed for these bee products for proof of efficacy. Future research should now focus on certain issues: (1) Comparing nutritional and medicinal values of bee products from different botanical and geographical origins would provide evidence on whether differences exist among these types. (2) Studies on whether monofloral BP and BB being of premium value, as in the case of Manuka and Sidr honey, compared to multifloral honey are recommended (3) Standardization of bee products on the basis of their nutrients is recommended. (4) Pharmacokinetics and bioavailability studies are recommended to capitalize more on biological value of bee products especially BP. (5) Nanoformulations as a novel drug delivery system is recommended to enhance the stability and bioavailability of bee products. Ultimately, considering the functional food industry with both nutritional and health value, bee products can potentially fulfill the demand for ingredients to produce food supplements with tremendous health value that has yet to be fully exploited.

## Figures and Tables

**Figure 1 molecules-28-00715-f001:**
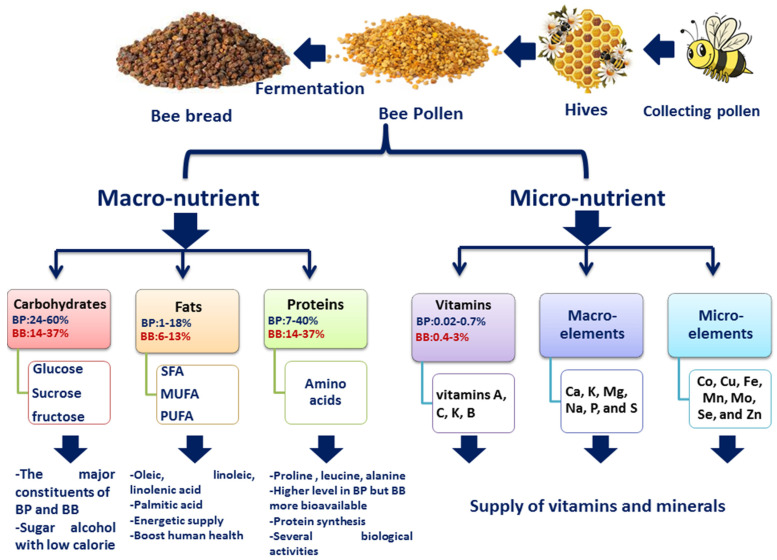
Schematic presentation of the collection of BP and formation of BB, and the macro- and micronutrients identified in both bee products, such as carbohydrates, fats, proteins, vitamins, macro-, and microelements.

**Figure 2 molecules-28-00715-f002:**
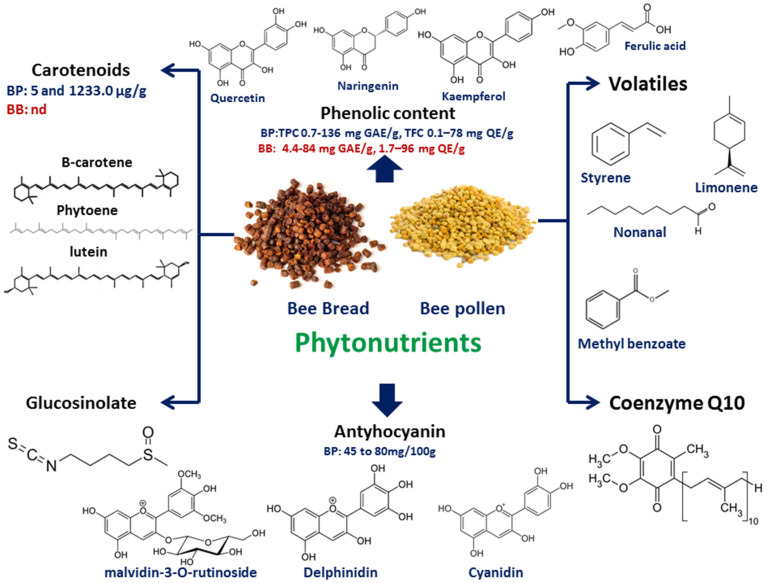
Major phytonutrients identified in both BP and BB, including phenolics, anthocyanins, carotenoids, glucosinolates, coenzyme Q10, and volatiles.

**Figure 3 molecules-28-00715-f003:**
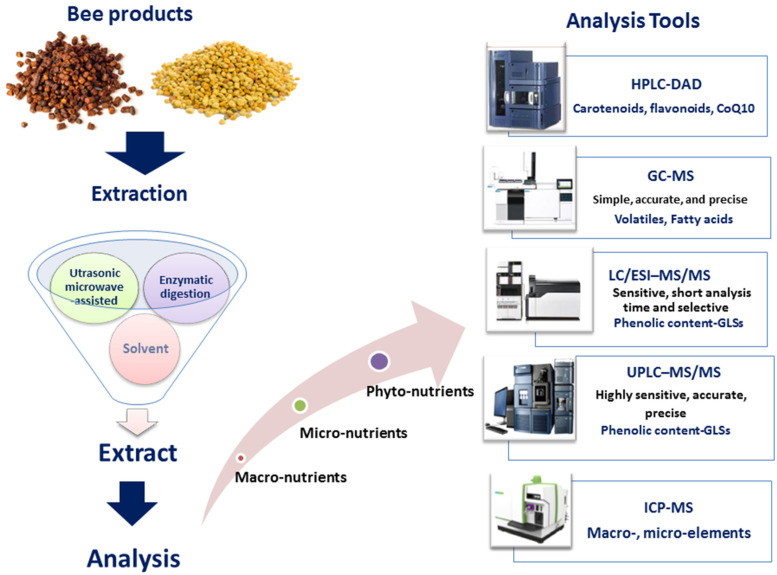
Methods of extraction and analysis of BP and BB macro-, micro- and phytoconstituents.

**Figure 4 molecules-28-00715-f004:**
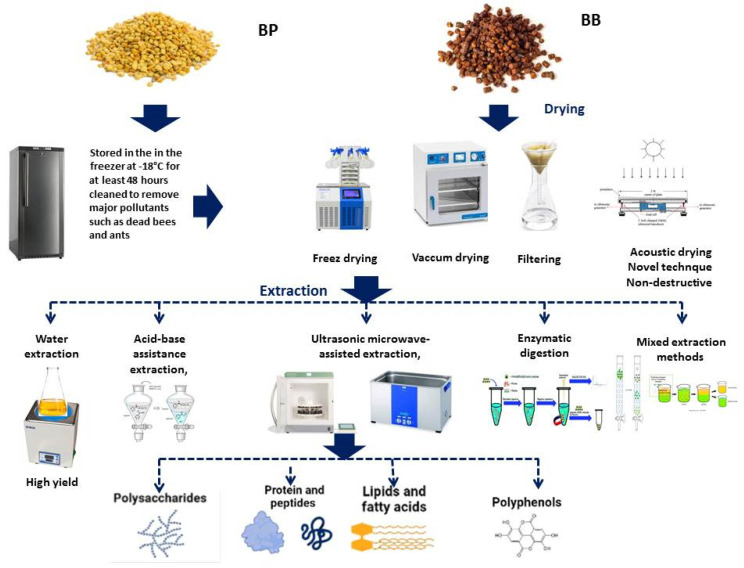
Schematic presentation of the stages of drying and extraction of bee products prior to the isolation of their bioactive metabolites.

**Table 1 molecules-28-00715-t001:** Different analytical methodologies used for the analysis of BP and BB samples for macro-, micro-, and phytonutrients.

Sample	Method of Analysis	Results	Advantages	Reference
Macronutrients				
BP samples (5 samples)	HPTLC and HPLEC-PAD	Sugars detected in all samples as stachyose, sucrose, glucose, galactose, and fructose		[4]
BP from three different Malaysian species	GC-MS	23 sugars were identified and mannitol is major in all species followed by fructose, xylitol, glucopyranose, ribitol, myo-inositol, and inositol.		[16]
Brazilian pollen samples (14 samples)	GC-FID	Fatty acids were oleic acid, linoleic acid, and arachidic acid, respectively.		[17]
Chinese monofloral BP species (20 samples)	GC-MS/SIM	14 saturated fatty acids, 4 MUFA, and 6 PUFA were identified: α-linolenic acid, nervonic acid, linoleic acid, arachidonic acid, and palmitic acid.	lower detection limits	[18]
BB (8 samples)	GC-MS	37 fatty acids were identified and the most abundant were palmitic acid, α-linoleic acid, oleic acid, stearic acid, and linolenic acid.		[19]
Tuscan BP species (5 samples)	GC-MS	18 fatty acids were detected: tridecanoic, palmitic, margaric, stearic, linolenic, and α-linolenic acid.		[20]
BP from 6 different sources from Saudi Arabia	HPLC	Oleic acid, palmitic acid, linolenic acid, stearic acid, and linoleic acid were detected		[21]
Lithuanian BP and BP	GC-FID	A total SFA range of 0.51–1.82% and total USFA range of 0.03–3.7% in bee samples.	Simple and sensitive	[22]
Malaysian BP (Three samples)	GC-MS	7 fatty acids were identified and α-linolenic acid was the highest level followed by linoleic acid with 0.07–1.11% in all bee samples.		[16]
BP from *Cocos nucifera* L.	GC-MS	Palmitic acid, oleic acid, and behenic acid were identified		[23]
Chinese monofloral BP (12 samples)	GC-FID	α-Linolenic acid, palmitic acid, oleic acid, linoleic acid, and stearic acid were detected at high level.		[24]
BP (Seven samples) from *F. esculentum* L.	GC-MS	USFAs accounted for 69.14%, with (9,2,15)-octadecatrienoic acid being the major one, detected in 36.25% of the total fatty acids		[25]
Turkish BP and BB samples (10 samples)	GC–MS	22 fatty acids were detected, mainly palmitic, stearic, oleic, linoleic, eicosenoic, and linoleic acid.		[26]
BP rape		19 amino acids were identified in rape BP samples	sensitive	[27]
Spanish bee pollen from *Cistus ladanifer* L. and *Echium plantagineum* L. (32 samples)	HPLC-fluorimetric	22 free amino acids were identified, mainly proline, leucine, alanine, phenylalanine, histidine, serine, and glutamine.		[28]
BP and BB (Five samples)	LC-MS/MS	42 free amino acids were presented, while the major amino acid was L-proline followed by asparagine, aspartic acid, and a high content of γ-aminobutyric acid.		[29]
BP ethanolic extract (from three Malaysian species)	GC–MS	23 sugars were detected in three species. L-alanine, L-valine, and L-asparagine were found in *Trigona apicalis* while L-proline and trimethylsiloxy proline were detected in *T. thoracica*		[16]
Chinese monofloral BP (Twelve samples)	GC-FID	18 amino acids were detected; Proline, glutamic acid, and aspartic were the major amino acids.		[24]
BP from *Rhododenron ponticum* L.	LC-MS	13 amino acids were detected and the main were L-tryptophan, L-glutamic acid, L-aspartic acid, and 3-amino isobutyric acid		[30]
Micronutrients				
BP *Rhododendron ponticum* L.	HPLC-FLD and HPLC-UV	Vitamins C, B5, B2, E, B6, B1, A-β-carotene, B12, A-retinol, B7, and K2 were identified		[30]
Polish BP samples	ICP-MS and ICP-OES	K, P, and S were detected at high levels and microelements such as Fe, Zn and Mn were detected.		[31]
Bee pollen (10 samples)	LS-FAAS with UAE	Ca, Cu, Fe, Mg, Mn, and Zn were detected.	Fast and simple	[32]
BP samples (144 samples)	NIR-PLS	Ca, Mg, Zn, P, and K were detected.	Fast	[33]
Chinese monofloral BP (12 samples)	GC-FID	Macro- and microelements such as P, K, Ca, Mg, and Na were detected.		[24]
BP and BB samples (10 samples)	ICP-MS	42 elements in all of the BP and BB samples were detected and the main elements were K, P, Mg, Ca, and Si.		[26]
BP (Eighteen samples)	ICP-MS	Co and Sr, and Ca and Mg were detected		[34]
BP sample of *Rhododenron ponticum* L.	ICP-MS	K, P, and Ca, and Mg, Cu, Fe, Mn, Na, Zn, and Ni were detected		[30]
Volatile compounds				
HoneyBP (Fourteen samples)	SPME-GC-MS	42 volatiles were identified, such as nonanal dodecane and tridecane.		[35]
Three different polyfloral Lithuanian BP (Three samples)	SPME-GC-MS	Styrene was the most abundant component in all samples.		[36]
Ground BP and its aqueous solution.	HS-SPME/GC–MS	25 volatiles were identified in ground BP and 22 from aqueous solution.		[37]
Stingless BP samples (genus *Scaptotrigona* sp.)	GC-MS	The most abundant volatile compounds identified were kaur-16-ene, methyl cinnamate, benzyl acetate, methyl benzoate, methyl hidrocinnamate, and ethyl phenylacetate.		[38]
Coenzyme Q10				
Rape BP, tea BP, apricot BP and mixed BP (11 commercial samples)	HPLC-DAD	Detection of CoQ10 in bee pollen		[1]
BB commercial sample	LC/MS-MS.	Coenzyme Q10 amounted to 11.5 μg/g and α-tocopherol was detected at a level of 80 μg/g	Sensitive, short analysis time and selective	[39]
Carotenoids				
BP samples	(RRLC)-UV–Vis spectrophotometry	11 Carotenoids were identified, mainly *α*-tocopherol and Phytoene.		[40]
Turkish chestnut BP (Sixteen samples)	HPLC-DAD	5 carotenoids were identified as lutein, zeaxanthin, β-cryptoxanthin, a-carotene, and *β*-carotene.		[41]
Glucosinolates				
BP (Five samples)	LC–MS/MS	Determination of Sulforaphane	Sensitive, short analysis time and selective	[42]
Spanish BP samples (Forty nine samples)	UPLC-Q-TOF/MS	glucosinolate was detected at a range 34–9806 μg/kg		[43]
Phenolic compounds				
Dark blue BP (Five samples)	HPLC-MS	8 anthocyanins were identified and petunidin-3-O-rutinoside was the major compound detected.		[2]
BP of *Fuchsia extorticata* L. and blue pollen	HPLC	Delphindin, petunidin, and malvidin-3-O-glucosides, and delphinidin-3-O-glucosides were detected.		[7]
BP sample	LC-MS	9 polyphenols were identified including ferulic acid, o-, *p*-coumaric acid, quercetin, cinnamic acid, naringenin, hesperitin, and kaempferol	higher sensitivity and shorter analysis time	[44]
BP methanol extract sample	HPLC	3, 4-Dimethoxycinnamic acid was detected at higher levels followed by gallic acid, catechin, and quercitin.		[45]
Commercial BP sample	SFC	9 polyphenolic compounds detected and the most abundant were catechin, quercetin, *p*-coumaric acid, and cinnamic acid	high resolution and short analysis time with low usage of solvents	[46]
Indian BP (35 samples)	UHPLC-DAD-MS/MS	60 compounds were identified and the most abundant compounds were catechin, rutin, quercetin, luteolin, kaempferol, and apigenin		[47]
Turkish Chestnut BP (16 samples)	HPLC-DAD	29 phenolic acids were identified and the major phenolic compounds were luteolin, hyperoside, vitexin, trans-chalcone, rosmarinic acid, pinocembrin, and chrysin		[41].
Mixture of Turkish pollen samples	LC-MS/MS	23 phenolic compounds were identified and the most abundant was rutin followed by kaempferol, quercetin, myricetin, and p-coumaric acid.	Sensitive, short analysis time and selective	[48]
BP and BB (5 fresh samples)	LC-MS/MS	23 phenolic compounds were identified. Rutin was found in higher concentrations in both BP and BB. Other phenolic compounds were detected at much higher levels in BP over BB, such as caffeic acid, ethyl gallate, trans-ferulic acid and myricetin, protocatechuic acid, *p*-coumaric acid, quercetin, isorhamnetin, luteolin, salicylic acid, chlorogenic acid, 2, 5-dihydroxybenzoic acid, kaempferol, and gallic acid.	Sensitive, short analysis time and selective	[29]
Honey BP (Fourteen samples)	HPLC-ECD	11 phenolic acid and flavonoids compounds were identified, such as 2-hydroxycinnamic acid, rutin, and quercetin.		[35]
BP from *Cocos nucifera* L.	HPLC-DAD-ESI-MS/MS	13 phenolic compounds were identified, such as isorhamnetin-3-O-(2″-O-rhamnosyl) glucoside, isorhamnetin-3-O-(2″,3″-O-dirhamnosyl) glucoside, and isorhamnetin-di-3,7-O-glucoside, as well as hydroxycinnamic acid amide derivatives.		[23]
BP of *Rhododendron ponticum* L.	LC–MS/MS	33 phenolic compounds were identified, with myricetin at the highest level followed by epicatechin, catechin and tyrosol.	Sensitive, short analysis time and selective	[2]

**Table 2 molecules-28-00715-t002:** Reported amino acids levels from bee pollen and bee bread (mg/g).

Sample	Arg	Asp	Ser	Glu	Thr	Gly	Ala	GABA	Pro	Val	Phe	Ile	Leu	(Cys)2	His	Lys	Try	Tyr
Rape bee pollen (mg/g) Xinjiang	7.27	10.6	8.26	9.5	5.83	6.28	6.5	0.34	11.4	6.4	4.89	4.96	9.28	-	2.9	9.85		3.86
Rape bee pollen (mg/g) Qinghai	6.45	14.1	7.97	11.85	6.82	6.46	7.13	0.16	12.17	9.07	5.25	5.02	9.86	0.56	3.89	10.36		4.27
Bee-pollen free AA (mg/g)	2.48	0.4	0.6	0.25	0.25	0.21	0.82	0.35	20.27	0.21	0.75	0.51	0.91	-	0.74	0.26		0.32
Bee-pollen total AA (mg/g)	5.03	15.1	2.74	17.88	4.17	6.4	10.86	-	22.88	7.26	9.62	9.22	10.81	-	6.84	10.97		7.43
Bee-pollen (mg/g)		7.15		7.79		1.98		5.33	2.29	2.23	2.71	1.59	2.09		1.22		8.05	1.25

**Table 3 molecules-28-00715-t003:** Macro- and microelements detected in bee bread and bee pollen.

Sample/ Elements	K	P	S	Fe	Zn	Mn	Ca	Cu	Mg	Na	Al	Si	Ni	Ref.
Polish bee pollen	4233.3 mg/kg	4050.0 mg/kg	2383.3 mg/kg	114.5 mg/kg	31.3 mg/kg	25.0 mg/kg	--	--	--	--	--	--	--	[31]
Bee pollen	--	--	--	2.2–3.9%	2.1–3.1%	2.0–2.8%	3.0–4.4%	2.2–3.5%	1.8–3.5%	--	--	--	--	[32]
Bee pollen (Brazilian)	3182 to 7376 mg/kg	3257 to 6886 mg/kg	--	--	38 to 76 mg/kg	--	1346 to 3724 mg/kg	--	915 to 1744 mg/kg	--	--	--	--	[33]
Chinese monofloral bee pollen	5324 mg/kg	5946 mg/kg	--	119.3 mg/kg	45.10 mg/kg	70.23 mg/kg	2068 mg/kg	17.35 mg/kg	1449 mg/kg	483.4 mg/kg,	129.3 mg/kg	--	--	[24]
BP and BB samples	5429.27–8994.25 mg/kg	4221.86–5948.96 mg/kg	--	--	--	--	189.69–447.13 mg/kg	--	1033.72 mg/kg	--	--	47–537.97 mg/kg	--	[26]
bee pollen sample of *Rhododenron ponticum* L.	3002.084 mg/kg	9145.125 mg/kg	--	47.007 mg/kg	28.760 mg/kg	25.629 mg/kg	459.507 mg/kg	13.496 mg/kg	1113.509 mg/kg	113.707 mg/kg	--	--	3.764 mg/kg	[30]

## Data Availability

Data are available within the article.

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
