# Peer review of "Bee Pollen and Bread as a Super-Food: A Comparative Review of Their Metabolome Composition and Quality Assessment in the Context of Best Recovery Conditions"

_molecules, 2023, doi:10.3390/molecules28020715_

Round 1

Reviewer 1 Report

This manuscript “Bee pollen and bread as a super-food: A comparative review of their metabolome composition and quality assessment in the context of best recovery conditions” is about bee pollen and bread for their health endorsing properties. This is an interesting from the scientific and market point of view. The given information is in this manuscript is useful for researchers and academia and article would be great contribution in discipline of pharmacology, eastern medicine, and nutrition and related discipline.   The room for improvement is always there and I have suggested some minor revisions. I can extend my services to further review the incorporation of the corrections in article again.

·         Abstract- is there any technical difference between fermented and functional foods? : Recently, functional and fermented foods have been a subject of great interest by dietetics 11 owing not only to their nutritional value but rather their myriad of health benefits.

·         L-12-15 Reference in abstract???????????? Is it as per the journal format to insert reference in abstract section? Moreover, the 12 growth in consumers' demands on such valuable foods makes a development in the production and hence tools of quality and nutrients assessment by introducing novel technologies in extraction, analysis, and processing. Bee products viz, pollen (BP) and bread [1] are normally harvested from 15 the flowering plants by the aid of bees

·         Need clarity -The fermentation process in hives makes BB more valuable 16 and bioavailable than BP????

·         L-48-49 Please recheck values-The total phenolic content (TPC) 48 of both BP and BB collected from different locations varies between 10.5–24.62 mgGAE/g 49 [12] and 2.5–37.15 mgGAE/g, respectively, and total flavonoid content (TFC) are reported 50 to be at 1.9–4.5 mg QE/g, respectively

·         L-124-125 the sentence is creating no sense- Results revealed for the presence of 37 fatty acids including pal- 124 mitic acid (22.3–38.7%), α-linoleic acid (6.3–37%), oleic acid (3.9–21.2 %), stearic acid (1.3 125 to 6.3 %) and linolenic Acid (0.2–40.7 %) as most abundant fatty acids.

·         Incomplete sentence-L-222-223 The use of LC-MS/MS to profile AAs in five fresh samples of BP and bee bread???????????????

·         L-299-300 the sentence again is incomplete-Another simple analytical method was modified for the 299 Molecules 2022, 27, x FOR PEER REVIEW 7 of 25 detection of 39 major, minor, trace and rare earth elements using small mass samples of 300 BP using ICP-

·         Who did this? Carotenoids composition of bee pollen samples (classified into twelve groups) were 360 analyzed using rapid resolution liquid chromatography (RRLC) coupled to UV–Vis spec- 361 trophotometry and the color characteristics were determined using Digital Image Anal- 362 ysis (DIA)[58].

·         L-373-374 Creating confusion?????????????????Comparison of carotenoid levels in in BP 373 to that of carrot later as major source of provitamin A, revealed the abundance of carote- 374 noid at higher levels posing it as a potential essential carotenoids source [59].

·         Creating no understanding- Numerous lipid-based nano-encapsulation methods have been invented to boost the antioxidant action of the individual constituents by increasing their solubility and bioavailability [97] and making them easier to target, place, and absorb [98]

·         Cite the following latest articles in under different sections to improve the overall qulaiy of the manuscript

o   Sevin, Sedat, et al. "Concentration of essential and non-essential elements and carcinogenic/non-carcinogenic health risk assessment of commercial bee pollens from Turkey." Journal of Trace Elements in Medicine and Biology 75 (2023): 127104.Please avoid repetition-

o   El-Seedi, H. R., Eid, N., Abd El-Wahed, A. A., Rateb, M. E., Afifi, H. S., Algethami, A. F., ... & Khalifa, S. A. (2022). Honey Bee Products: Preclinical and Clinical Studies of Their Anti-inflammatory and Immunomodulatory Properties. Frontiers in Nutrition8, 761267.

o   Liolios, V., Tananaki, C., Kanelis, D., & Rodopoulou, M. A. (2022). The microbiological quality of fresh bee pollen during the harvesting process. Journal of Apicultural Research, 1-11.

o   Sevin, S., Tutun, H., Yipel, M., Aluç, Y., & Ekici, H. (2023). Concentration of essential and non-essential elements and carcinogenic/non-carcinogenic health risk assessment of commercial bee pollens from Turkey. Journal of Trace Elements in Medicine and Biology75, 127104.

o   Sharma, A., Pant, K., Brar, D. S., Thakur, A., & Nanda, V. (2022). A review on Api-products: current scenario of potential contaminants and their food safety concerns. Food Control, 109499.

·         Please check reference style throughout MS

·         Italic all the scientific names,

·         Remove grammatical mistakes

·         Need to rewrite the conclusion

Author Response

Please find our response in the attachment. 

Reviewer 2 Report

Dear Authors,

Please find my suggestions below.

This is interesting review related to the metabolome composition and quality of bee pollen and bread.

There is different fonts and size in manuscript. Choose one type of font and one size for all manuscript.

The sentence which starts in line 89 and ends in line 95 is too long.  Make two sentences.

The sentence which starts in line 138 and ends in line 140 is too long.  Make two sentences.

In the subsection 2.3 first four sentences were started on the same way. Correct it.

In the line 225 small letter I in in. Correct it.

The sentence which starts in line 292 and ends in line 296 is too long.  Make two sentences.

In the subsection 4.5 highlight that phenolic acids and phenolic compounds showed ability to activate enzymes of antioxidant protection in cells, which can prevent oxidative stress damage in cells.  Kindly consider to cite Agronomy, 11(7), (2021) 1414.

Wish you all the best in future work,

Author Response

(The authors gave the same response as above.)

Reviewer 3 Report

This is a review of bee pollen and bee bread composition. The aim of the review is of interest for readers because it aims to compare bee pollen and bee bread composition and the influence of processing and extracting techniques. However, in my opinion, the authors do not reach their purpose. The authors used many papers to write this review, but the information extracted from each of them is not processed adequately. The document needs a deep and robust revision before publication.

Most of the sections are merely a description of the data obtained in different works presented in a consecutive way. There is no analysis of these data, in order to give the reader a clear idea of the abundance/values/percentages of the different compounds in the products (bee pollen/bee bread).

For example, in fats section is described first the results of 14 brazilian pollen samples, then 20 chinese monofloral pollen, then another study about bee bread and others, but it is not presented clearly what are the fatty acids and their abundance in bee pollen and bee bread. My suggestion for the whole document is to merge the information obtained in the different papers, to prepare different tables unifying values of the different parameters and then rewrite the document in a more comprehensive way. Moreover, some minor tips should be considered. For example, some of the scientific names are included with the authorities, other no, in other cases the genus is abbreviated when it was not mentioned before.

Another suggestion is to change the figures by tables detailing the values of bee pollen and bee bread composition found in references.

Table 2 and Table 3 should be also re-do to be more useful and understandable.

The font size changes along the text. There are some mistakes in the text.

Author Response

(The authors gave the same response as above.)

Round 2

Reviewer 3 Report

The authors made a revision, but some weakness still persists. In my opinion, some of the sections should be reduced to facilitate comprehension, i.e. Fats section. This section includes first the health benefits of Fatty acids (it can be summarized in a single sentence) then results about bee pollen (Brazilian and Chinese), later results on bee bread (lines 125-131) and again results on bee pollen (Tuscanian and other). For readers, the section is really hard to follow. The same occurs with values sometimes expressed in percentage and others in mg/kg. Maybe it is possible to express data in the same units. This would be very helpful, also in table 3. The same occurs in other sections of the work.

Regarding tables 2 and 3, if columns and rows are transposed and the reference is added in the first column, the tables will be more suitable for publishing.

Table 2, please correct mg/gm by mg/g
